# Differences in the long-term course of post-COVID-19 symptoms in adults and children across epidemic periods: A retrospective cohort study in Japan, 2020–2024

Aya Sugiyama[1]*, Toshiro Takafuta[2], Kanon Abe[3], Yayoi Yoshinaga[1], Ko Ko[1], Tomoki Sato[2,4], Tomoyuki Akita[1], Masao Kuwabara[5], Shingo Fukuma[1], Junko Tanaka[1]

1 Department of Epidemiology Disease Control and Prevention, Graduate School of Biomedical and Health Sciences, Hiroshima University, Hiroshima, Japan, 2 Hiroshima City Funairi Citizens Hospital, Hiroshima, Japan, 3 Department of Preventive Gerontology, Center for Gerontology and Social Science, National Center for Geriatrics and Gerontology, Aichi, Japan, 4 Department of Pediatrics, Hiroshima City Funairi Citizens Hospital, Hiroshima, Japan, 5 Hiroshima Prefecture Center for Disease Control and Prevention, Hiroshima, Japan

* aya-sugiyama@hiroshima-u.ac.jp

## Abstract

### Background

The prevalence of post-COVID-19 symptoms has been reported to decline since the Omicron variant became predominant. However, differences in their long-term course across epidemic periods and between adults and children, including recent Omicron sublineages, remain insufficiently understood.

### Methods

We extended a previously reported retrospective cohort by conducting follow-up and an additional survey in Hiroshima, Japan. The study included 2,689 individuals diagnosed with COVID-19 between March 2020 and June 2024 (1,524 adults and 1,165 children). A self-administered questionnaire captured the presence and duration of 13 symptoms. Interval-censored survival analysis estimated prevalence over time, and proportional hazards models evaluated factors associated with symptom resolution.

### Results

At six months, the estimated prevalence in adults was highest during the Delta period (47%) and lower during Omicron-2022 (23%) and Omicron-2024 (21%). In children, prevalence remained about one-quarter to one-third that of adults, with no notable differences between Omicron sublineages. At two years, persistent symptoms were reported by about 20% of adults infected before Omicron and 10% during Omicron periods, compared with 4.1% and 1.9% of children infected during the Delta and Omicron-2022 periods. Symptoms persisting beyond two years showed little further

**Data availability statement:** Individual-level data cannot be shared, as participants did not consent to external release and the IRB-approved protocol prohibits it. All aggregated data underlying the results and the full analysis codes are openly available in Zenodo (DOI: https://doi.org/10.5281/zeno-do.17375257), which is linked to the GitHub repository (https://github.com/Aya-Sugiyama/longcovid-epiperiods-japan-2020-2024).

**Funding:** This study was supported by: JT: Japan Agency for Medical Research and Development (AMED) under Grant Numbers JP20fk0108453 and JP21fk0108550 (URL: https://www.amed.go.jp) AS: Japan Agency for Medical Research and Development (AMED) under Grant Number JP24fk0108706 (URL:https://www.amed.go.jp) JT: Hiroshima Prefecture Government-academia collaboration project funding The sponsors or funders had no role in the study design, data collection and analysis, decision to publish, or preparation of the manuscript.

**Competing interests:** The authors declare no competing interests.

resolution, though in children they did not interfere with daily activities. In the Cox model, resolution was slower during the Delta period (HR 0.79) and faster during Omicron-2022 (HR 1.24) and Omicron-2024 (HR 1.30). Younger age, particularly ≤12 years, was strongly associated with faster recovery.

## Conclusion

The long-term course of post-COVID-19 symptoms differed across epidemic periods and age groups. The risk was highest during Delta and lower among children and those infected during Omicron waves, yet some individuals experienced symptoms for over two years. Long-term follow-up and social support remain crucial to mitigate the burden of post-COVID-19 condition.

## Introduction

COVID-19, caused by infection with severe acute respiratory syndrome coronavirus 2 (SARS-CoV-2), emerged at the end of 2019 and triggered a global pandemic. The disease can result in long-term health consequences collectively referred to as post-COVID-19 condition (PCC), or long COVID, and since the onset of the pandemic, millions of people worldwide have been suffering from persistent symptoms [1–3]. Long COVID continues to be recognized as one of the most pressing global public health challenges [1,4]. According to the World Health Organization, symptoms generally improve over time, typically resolving within 4–9 months [1]. However, global estimates from 2022 suggest that approximately 15% of patients still experience symptoms 12 months after onset [1]. More recently, as the Omicron variant has become predominant, the frequency of long COVID has been reported to decline [5–8]. Nevertheless, evidence on the long-term course beyond two years, as well as on post-COVID-19 symptoms associated with Omicron sublineages, remains limited [9,10]. In addition, although children have been reported to experience long COVID less frequently and with milder symptoms than adults [11], few studies have directly compared adults and children under the same study conditions. In this manuscript, we primarily use the term "post-COVID-19 symptoms" to describe persistent symptoms following SARS-CoV-2 infection, while the terms "long COVID" and "post-COVID-19 condition" are used when referring to established definitions or prior literature.

We previously reported the long-term course of persistent post-COVID-19 symptoms among 2,421 individuals (1,391 adults and 1,030 children) who consented to participate out of 6,551 patients diagnosed with COVID-19 between March 2020 and July 2022 at a collaborating medical institution in Hiroshima Prefecture, including both outpatients and inpatients across all ages [12]. However, in that study, the follow-up period for patients infected with the Omicron variant, which became predominant after 2022, was less than one year, leaving their longer-term outcomes unclear. In addition, the study population was limited to those infected up to July 2022, and thus the characteristics of symptoms associated with subsequently emerging Omicron sublineages could not be evaluated. To address these gaps, we conducted follow-up

and additional surveys to examine how the prevalence, duration, and characteristics of post-COVID-19 symptoms varied by epidemic periods (wild-type, Alpha, Delta, Omicron-2022, and Omicron-2024). A key aim of the present study was also to clarify differences in the long-term course of post-COVID-19 symptoms between adults and children. Given that the incidence of long COVID has been reported to be lower in children than in adults [9,12], we analyzed adults and children separately in the present study.

## Materials and methods

### Study design and participants

This study was a follow-up and additional survey based on our previously conducted retrospective cohort study [12]. In the earlier study, all 6,551 patients (3,748 adults and 2,803 children) diagnosed with COVID-19 between March 2020 and July 2022 at a designated Class II infectious disease medical institution in Hiroshima Prefecture, which also serves as a core hospital for pediatric emergency care, were invited to participate. A survey was conducted between November 2022 and March 2023, and responses regarding persistent post-COVID-19 symptoms were obtained from 2,421 individuals (1,391 adults and 1,030 children). Details of the study design, survey methods, and participant characteristics have been reported previously [12]. The anonymized dataset used in the present study was constructed by the research team as described in our previous report [12]. For the current analysis, the dataset was accessed for research purposes in July 2024. Because all data had been anonymized at the time of construction, the authors had no access to any personally identifiable information.

In the present study, the analytic sample comprised the 2,421 respondents from the previous survey together with 761 newly diagnosed cases (382 adults and 379 children) identified between January and June 2024. Among participants in the previous survey (November 2022–March 2023), 466 individuals (403 adults and 63 children) who still reported post-COVID-19 symptoms were followed up to collect additional information. For participants in the follow-up survey of the original cohort, respondents were explicitly instructed to report only symptoms persisting from the previously surveyed infection episode and not to include symptoms newly developed after subsequent infections. The current survey was conducted between December 2024 and March 2025, and the data obtained were integrated into the database established in the previous study for analysis.

### Data collection and measures

A self-administered questionnaire was mailed to participants, asking about the presence, type, and duration of self-reported post-COVID-19 symptoms. Thirteen representative symptoms were assessed: fatigue, cough, shortness of breath, sleep disorders, altered smell, altered taste, headache, chest pain, dizziness, hair loss, limb pain, memory issues, and difficulty concentrating. For each symptom, participants were also asked whether it interfered with daily life. Because precise recall of symptom duration was difficult, responses were collected in ranges of several months. For pediatric participants, questionnaire responses were obtained primarily through proxy responses provided by parents or legal guardians. Parents or guardians completed the survey based on their observations, with input from the child when appropriate.

Information on the severity of acute COVID-19 was extracted from medical records and classified into four categories according to the need for oxygen support: mild (no oxygen required), moderate (oxygen therapy required), severe (use of noninvasive mechanical ventilation), and critical (use of invasive mechanical ventilation).

Infection periods were defined based on genomic surveillance data in Hiroshima Prefecture [13–15] and categorized as follows: Wild-type period (March 2020 – February 2021), Alpha period (March – June 2021), Delta period (July – November 2021), Omicron-2022 period (December 2021 – July 2022), and Omicron-2024 period (January – June 2024). For clarity, these infection periods are hereafter referred to as epidemic periods throughout the manuscript.

## Outcome

The primary outcome of this study was the duration of post-COVID-19 symptoms. For adults and children separately and across epidemic periods, we evaluated [1] the proportion of individuals reporting any level of severity, defined as the presence of at least one post-COVID-19 symptom, to capture the overall burden of persistent symptoms, and [2] the proportion reporting symptoms that interfered with daily life.

Secondary outcomes were as follows:

1. Adjusted hazard ratios (HRs) for symptom resolution: HRs for the resolution of post-COVID-19 symptoms were estimated among study participants.

2. Symptom-specific prevalence: the prevalence of each of the 13 symptoms (fatigue, cough, shortness of breath, sleep disorders, altered smell, altered taste, headache, chest pain, dizziness, hair loss, limb pain, memory issues, and difficulty concentrating) was assessed at 3 and 12 months after infection, stratified by epidemic periods.

## Statistical analysis

Participant characteristics were summarized separately for adults and children. Age was summarized as mean (standard deviation) and median (interquartile range), and categorical variables as frequencies and percentages.

The prevalence of post-COVID-19 symptoms was estimated using interval-censored survival analysis with the Turnbull method [16]. Time since recovery from acute infection was used as the time scale, and survival curves were estimated for five epidemic periods (wild-type, Alpha, Delta, Omicron-2022, and Omicron-2024). Because the observation intervals differed across epidemic periods, common time points (0, 1, 2, 3, 6, 12, 24, 36, and 48 months) were set to facilitate comparison. At each of these time points, point estimates of the survival function S(t) and corresponding 95% confidence intervals (CIs) were calculated. Analyses were performed separately for adults and children, and further stratified by symptom severity (any symptom, symptoms interfering with daily life).

Hazard ratios (HRs) and 95% CIs for symptom resolution were estimated using proportional hazards models for interval-censored data. Covariates included age at infection, sex, severity of acute illness, and epidemic periods. These analyses were conducted using the icenReg package (version 2.0.16) in R (version 4.4.1; R Foundation for Statistical Computing, Vienna, Austria).

Symptom-specific prevalence was estimated at 3 and 12 months after infection using the Turnbull method, aligning survival curves as step functions at the predefined observation times. For each epidemic period, point estimates and 95% confidence intervals were reported for 13 symptoms: fatigue, cough, shortness of breath, sleep disorder, altered smell, altered taste, headache, chest pain, dizziness, hair loss, limb pain, memory issues, and difficulty concentrating. In this analysis, adults and children were combined because the number of pediatric cases in non-Omicron periods was insufficient for separate estimation. As a supplementary analysis, symptom-specific prevalence at 3 and 12 months was also estimated separately for adults and children, irrespective of epidemic period (S1 Fig).

All analyses were performed using R (version 4.4.1) and JMP® (version 14; SAS Institute Japan, Tokyo, Japan). A two-sided significance level of $p < 0.05$ was applied throughout.

All analytic code and aggregated data required to reproduce the results are openly available on GitHub (https://github.com/Aya-Sugiyama/longcovid-epiperiods-japan-2020-2024/tree/main), archived with DOI: https://doi.org/10.5281/zenodo.17375257.

## Ethics declarations

This study was approved by the Ethics Committee of Hiroshima University (Approval No. E-2122) and conducted according to the Helsinki Declaration. Furthermore, written informed consent was obtained from each patient before any study procedure. For children, informed consent was taken from their parents or guardians.

## Results

### Participant characteristics

Among the 2,421 participants from the previous survey, 466 individuals (403 adults and 63 children) who reported persistent post-COVID-19 symptoms as of November 2022 were invited for follow-up, and 228 responded (response rate: 48.9%). In addition, of the 761 individuals (382 adults and 379 children) newly diagnosed with COVID-19 between January and June 2024, 268 (137 adults and 131 children) responded (response rate: 35.2%). In total, the analytic cohort consisted of 2,689 individuals (1,524 adults and 1,165 children), combining the 2,421 participants from the previous survey and the 268 new respondents (Fig 1).

Participant characteristics are presented in Table 1. The mean age was 32.2±25.6 years, with a median of 31 years (interquartile range: 8–53, range: 0–95). Women accounted for 49.1% of the cohort. Regarding severity of acute infection, mild cases comprised 83.9% of adults and 98.1% of children. The distribution of epidemic periods differed by age group: 29.8% of adults were infected during the Omicron-2022 period, compared with 79.2% of children. The proportion of hospitalized patients was 54.0% among adults and 8.1% among children.

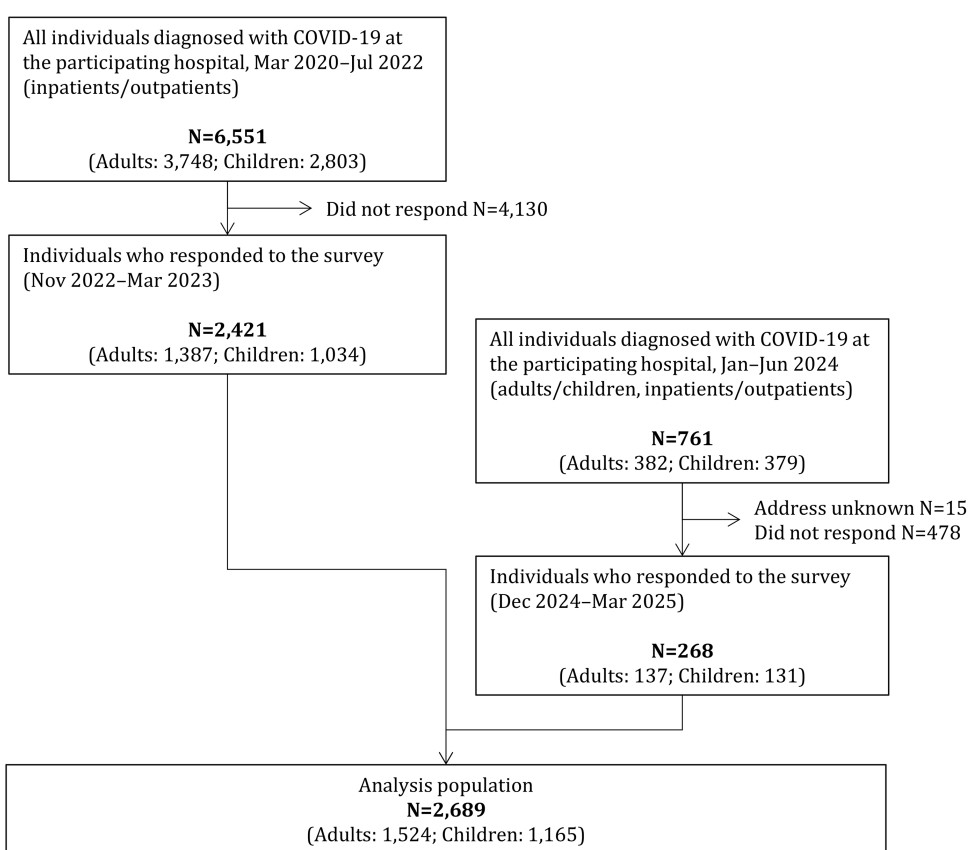

**Fig 1. Flow diagram of study population.** Of the 2,421 individuals who participated in the previous survey (Nov 2022–Mar 2023), 466 had persistent post-COVID-19 symptoms at that time. A follow-up survey (Dec 2024–Mar 2025) was conducted among these individuals, and responses were obtained from 228, whose information was subsequently updated.

**Table 1. Participants' characteristics.**

| | | Total | % | Adults | % | Children | % |
|---|---|---|---|---|---|---|---|
| **N** | | 2,689 | % | 1,524 | % | 1,165 | % |
| **Age at COVID-19 onset** | **Mean (SD), y** | 32.2(25.6) | | 51.7(16.4) | | 6.7(4.5) | |
| | **Median (IQR), y** | 31(8-53) | | 51(40-63) | | 6(3–10) | |
| **Sex** | **Female** | 1,321 | 49.1 | 805 | 52.8 | 516 | 44.3 |
| | **Male** | 1,368 | 50.9 | 719 | 47.2 | 649 | 55.7 |
| **Severity of COVID-19** | **Asymptomatic** | 37 | 1.4 | 19 | 1.2 | 18 | 1.5 |
| | **Mild** | 2,421 | 90.0 | 1,278 | 83.9 | 1,143 | 98.1 |
| | **Moderate** | 182 | 6.8 | 178 | 11.7 | 4 | 0.3 |
| | **Severe** | 46 | 1.7 | 46 | 3.0 | 0 | 0.0 |
| | **Critical** | 3 | 0.1 | 3 | 0.2 | 0 | 0.0 |
| **Epidemic periods** | **Wild-type period** | 406 | 15.1 | 379 | 24.9 | 27 | 2.3 |
| | **Alpha period** | 320 | 11.9 | 309 | 20.3 | 11 | 0.9 |
| | **Delta period** | 318 | 11.8 | 245 | 16.1 | 73 | 6.3 |
| | **Omicron period-2022** | 1,377 | 51.2 | 454 | 29.8 | 923 | 79.2 |
| | **Omicron period-2024** | 268 | 10.0 | 137 | 9.0 | 131 | 11.2 |
| **The locations for recovery at the time of infection** | **Home** | 1,508 | 56.1 | 507 | 33.3 | 1,001 | 85.9 |
| | **Quarantine hotel** | 264 | 9.8 | 194 | 12.7 | 70 | 6.0 |
| | **Hospital** | 917 | 34.1 | 823 | 54.0 | 94 | 8.1 |

Severity was classified into four categories based on the need for oxygen or ventilation support; mild (i.e., no need for supplemental oxygen), moderate (i.e., needed for supplemental oxygen), severe (i.e., non-IMV use), and critical (i.e., IMV use).

## Estimated prevalence of post-COVID-19 symptoms in Adults and Children by epidemic period

The distribution of intervals to symptom resolution among participants is presented in S1 Table. Using these data, interval-censored survival analysis with the Turnbull method was performed, and the estimated prevalence of post-COVID-19 symptoms over time was plotted by epidemic period separately for adults and children (Fig 2).

In adults, the estimated prevalence at 6 months after recovery from acute infection was 36.4% during the wild-type period, 38.7% during the Alpha period, 47.2% during the Delta period, 22.5% during the Omicron-2022 period, and 20.6% during the Omicron-2024 period (S2 Table). For symptoms interfering with daily life, the prevalence was highest during the Delta period (26.4%), followed by 14.6% in the wild-type period, 15.9% in the Alpha period, 10.8% in the Omicron-2022 period, and 11.7% in the Omicron-2024 period.

In children, the estimated prevalence at 6 months was consistently lower than in adults: 11.1% in the wild-type period, 11.4% in the Delta period, 5.6% in the Omicron-2022 period, and 6.4% in the Omicron-2024 period. For symptoms interfering with daily life, the corresponding prevalences were 3.7% (wild-type), 10.3% (Delta), 2.8% (Omicron-2022), and 1.5% (Omicron-2024).

At 2 years after infection, approximately 20% of adults infected during the wild-type, Alpha, or Delta periods and about 10% of those infected during the Omicron periods continued to report symptoms. At the same time point, persistent symptoms were observed in 4.1% of children infected during the Delta period and 1.9% during the Omicron-2022 period. Symptoms that persisted for more than 2 years showed little further resolution thereafter in both adults and children; however, in children, these symptoms did not interfere with daily activities and included fatigue, headache, dizziness, cough, difficulty concentrating, and sleep disorders.

**A) Adults (N=1,524)**

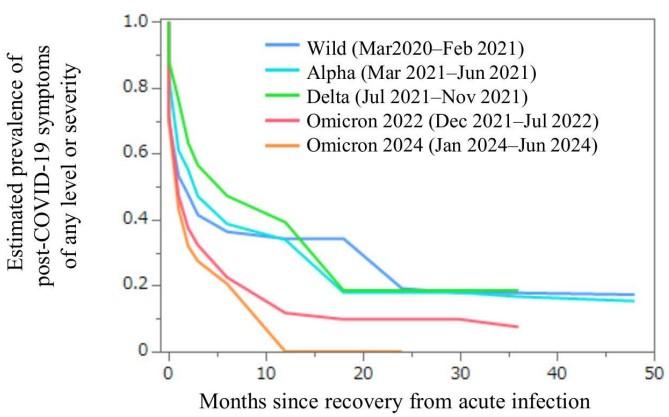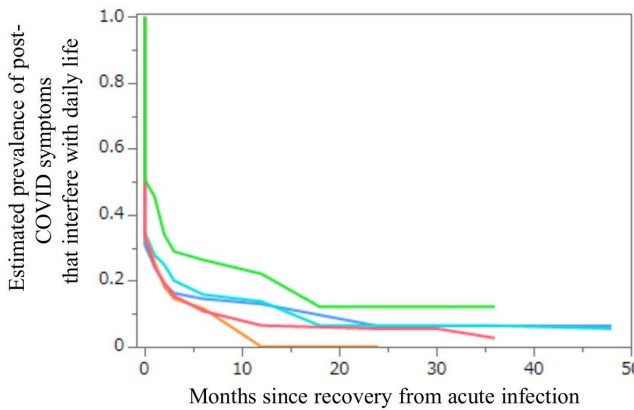

**B) Children(N=1,165)**

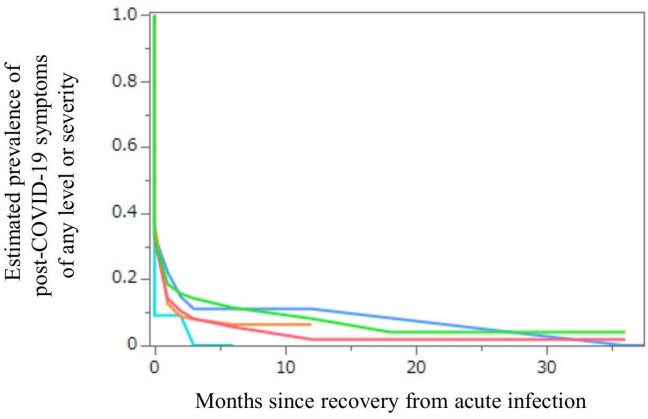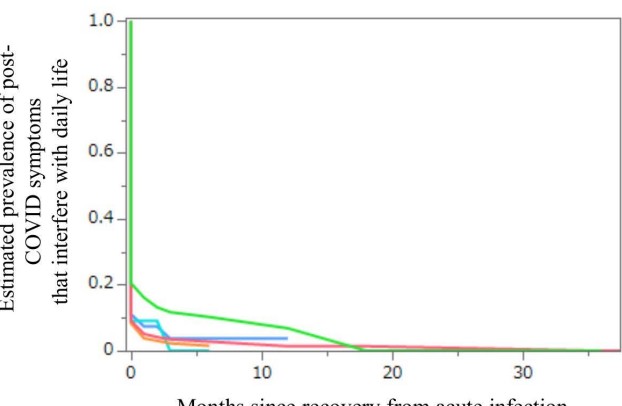

**Fig 2. Estimated prevalence of post-COVID-19 symptoms over time, stratified by epidemic periods, age group, and severity.** Survival analysis was conducted using the Turnbull method to estimate the prevalence of persistent post-COVID-19 symptoms. The x-axis indicates months since recovery from acute infection. Curves represent infection period groups: blue, Wild (Mar 2020–Feb 2021); light blue, Alpha (Mar 2021–Jun 2021); green, Delta (Jul 2021–Nov 2021); red, Omicron 2022 (Dec 2021–Jul 2022); and orange, Omicron 2024 (Jan 2024–Jun 2024). **A)** Adults; **B)** Children. Left panels: any level of symptom severity; right panels: symptoms interfering with daily life..

## Adjusted hazard ratios for resolution of post-COVID-19 symptoms

Adjusted hazard ratios (HRs) for symptom resolution are shown in Table 2. Symptom resolution was strongly associated with age. Using ages 0–12 years as the reference, adjusted HRs were 0.53 for ages 13–29 years, 0.38 for 30–49 years, 0.33 for 50–69 years, and 0.39 for ≥70 years, indicating that symptoms resolved more readily in younger individuals, particularly in children aged 12 years or younger. Compared with the wild-type period, participants infected during the Delta period were significantly less likely to experience symptom resolution (adjusted HR 0.79, 95% CI: 0.66–0.94). In contrast, resolution was significantly more likely during the Omicron-2022 period (adjusted HR 1.24, 95% CI: 1.06–1.44) and the Omicron-2024 period (adjusted HR 1.30, 95% CI: 1.07–1.57).

Other factors independently associated with delayed resolution of post-COVID-19 symptoms included moderate or severe acute disease severity, and female sex.

**Table 2. Adjusted hazard ratios (HR) for resolution of any post-COVID-19 symptoms among individuals with a history of COVID-19 infection.**

|  | Variable | HR | 95%CI | p |
|---|---|---|---|---|
| **Age** | 0-12 (ref) | 1.00 |  |  |
|  | 13-29 | 0.53 | 0.46-0.61 | <0.0001 |
|  | 30-49 | 0.38 | 0.33-0.44 | <0.0001 |
|  | 50-69 | 0.33 | 0.28-0.38 | <0.0001 |
|  | 70- | 0.39 | 0.33-0.46 | <0.0001 |
| **Sex** | Female (ref) | 1.00 |  |  |
|  | Male | 1.12 | 1.02-1.22 | 0.0140 |
| **Severity of COVID-19** | mild (ref) | 1.00 |  |  |
|  | moderate/severe | 0.77 | 0.66-0.91 | 0.0020 |
| **Epidemic periods** | Wild-type (ref) | 1.00 |  |  |
|  | Alpha | 0.96 | 0.81-1.13 | 0.5990 |
|  | Delta | 0.79 | 0.66-0.94 | 0.0100 |
|  | Omicron-2022 | 1.24 | 1.06-1.44 | 0.0060 |
|  | Omicron-2024 | 1.30 | 1.07-1.57 | 0.0080 |

Hazard ratios and 95% confidence intervals were estimated using a proportional hazards model for interval-censored data, adjusting for age at infection, sex, disease severity, and epidemic periods.

## Symptom-specific characteristics by epidemic period

Figure 3 shows the estimated prevalence of individual symptoms at 3 and 12 months after recovery from acute infection. Across epidemic periods, fatigue generally ranked high in prevalence Distinctive patterns were observed by epidemic period: during the Delta period, altered smell and altered taste were common, and this tendency persisted at 12 months. In contrast, although the overall prevalence of symptoms was lower during the Omicron periods, cough remained relatively common. No notable changes in the symptom profile were observed in the Omicron-2024 period. When comparing adults and children, the overall prevalence of post-COVID-19 symptoms was markedly lower in children; however, cough and difficulty concentrating were relatively more frequent among pediatric cases (S1 Fig).

## Discussion

In this study, we examined the prevalence, duration, and characteristics of post-COVID-19 symptoms among adults and children infected with COVID-19 between March 2020 and June 2024, stratified by epidemic periods (wild-type, Alpha, Delta, Omicron-2022, and Omicron-2024). This study also addressed the knowledge gap that few studies have directly compared adults and children under the same study conditions.

The main findings were as follows. The prevalence of post-COVID-19 symptoms was markedly lower in children than in adults across all epidemic periods, and, in this study, none of the children experienced symptoms interfering with daily life beyond two years after infection. The proportion of individuals with post-COVID-19 symptoms was highest during the Delta period and declined among those infected during the Omicron periods, with this tendency confirmed in the Omicron sublineages circulating in 2024. Adjusted hazard ratios for symptom resolution showed that children, particularly those aged ≤12 years, recovered more rapidly than older age groups. Compared with the wild-type period, resolution was significantly less likely in the Delta period (HR 0.79) and significantly more likely in the Omicron-2022 (HR 1.24) and Omicron-2024 (HR 1.30) periods.

Regarding duration, among adults, approximately 20% of pre-Omicron infections and about 10% of Omicron infections still had persistent symptoms two years after infection. In children, prevalence was substantially lower, with around 4% for pre-Omicron and 2% for Omicron infections. Symptoms persisting for more than two years rarely resolved thereafter. This

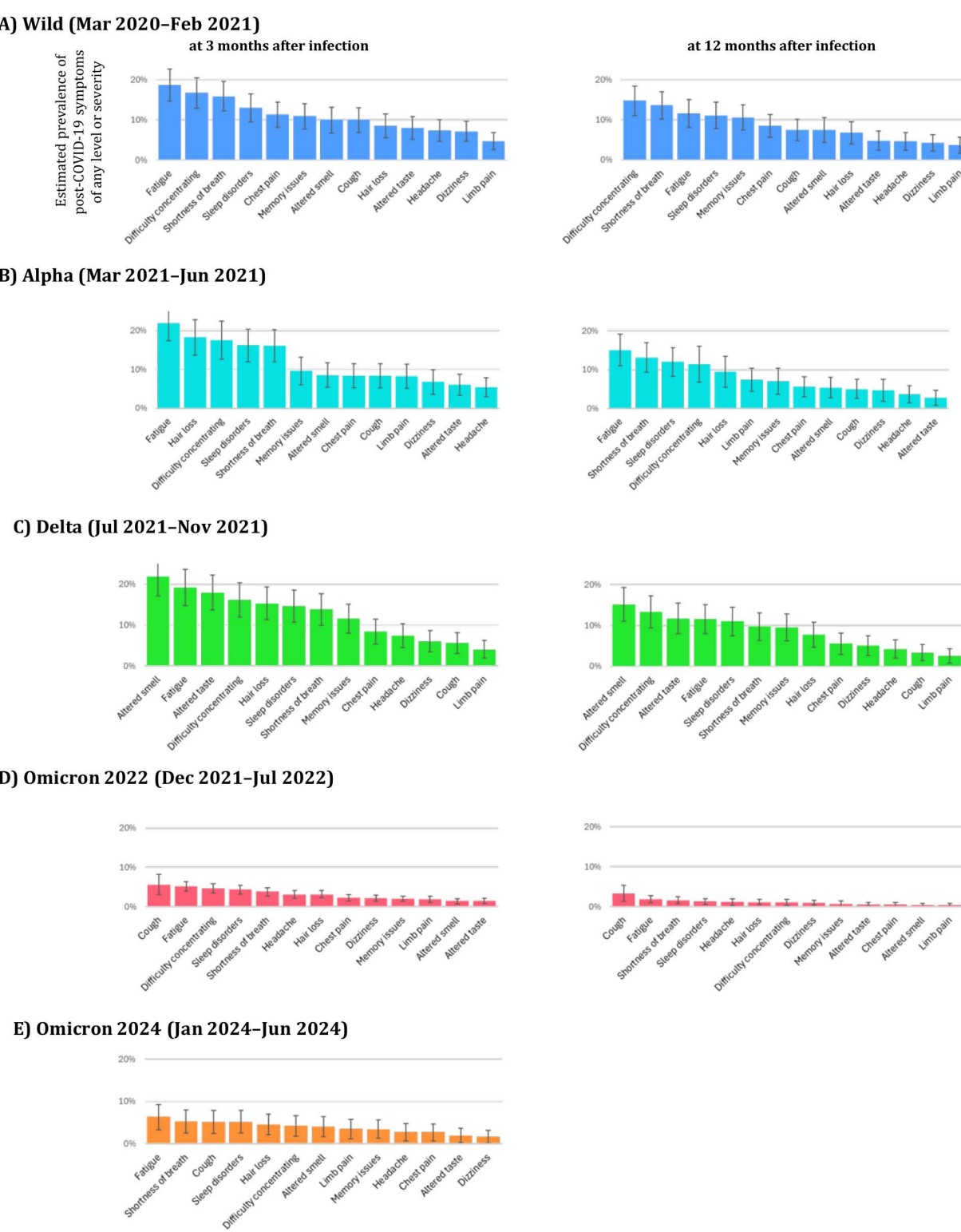

**Fig 3. Estimated prevalence of 13 post-COVID-19 symptoms at 3 and 12 months after infection, stratified by epidemic periods.** Prevalence was estimated at two common time points (3 and 12 months after infection) using the Turnbull method with step-function alignment of survival curves. Bars represent prevalence estimates with 95% confidence intervals. Symptoms on the x-axis are ordered by prevalence within each infection period group. Left panels: prevalence estimates at 3 months after acute infection; right panels: prevalence estimates at 12 months after acute infection. A total of 2,689 participants were included..

finding reflects the observed plateau in symptom resolution within the available follow-up period and does not preclude the possibility of further improvement beyond the observation window.

With respect to symptom type, altered smell and altered taste were more common during the Delta period, whereas no distinctive symptom pattern was observed for the Omicron-2024 period.

Although a decline in the frequency of post-COVID-19 symptoms since the emergence of the Omicron variant has already been reported, most previous studies compared only the Delta (or pre-Omicron) and Omicron periods [5,6,17,18]. Few studies have examined differences across multiple epidemic periods, including wild-type, Alpha, Delta, and Omicron. Furthermore, few have investigated these differences among children. In addition, evidence regarding Omicron sublineages remains scarce [7,10].

Before interpreting these differences, it should be noted that the comparisons across epidemic periods in this study represent period-based associations rather than direct causal effects of specific viral variants. Although the interval-censored survival curves were generated using a uniform analytic framework across all periods, they reflect underlying differences in participant composition, clinical severity, and sample size across epidemic periods. Importantly, however, the association between epidemic period and symptom resolution remained statistically significant after adjustment for relevant covariates in the Cox proportional hazards models.

In this study, we demonstrated that the risk of post-COVID-19 symptoms was significantly higher among individuals infected during the Delta period than in other periods, and this tendency was also observed among children. The Delta variant carries the P681R mutation in the spike protein and exhibits enhanced cell–cell fusion capacity, both of which have been shown in epidemiological studies and animal models to contribute to increased disease severity [19]. Our findings are consistent with the potential clinical implications of these virological features.

With regard to Omicron sublineages, a prospective community cohort study in the United Kingdom reported that, compared with early BA.1 infections (December 2021–March 2022), the prevalence of symptoms lasting more than two months was lower with BA.2 and BA.5 and was similar to that observed in other acute respiratory infections [10]. That study included infections up to March 2023, whereas our study extended the evidence by analyzing cases from January to June 2024, showing that the frequency, duration, and types of post-COVID-19 symptoms in more recent Omicron sublineages did not substantially differ from those observed during the Omicron period of 2022. According to genomic surveillance in Hiroshima Prefecture, Omicron-2022 corresponds to periods dominated by BA.1, BA.2, and BA.5, while Omicron-2024 corresponds to EG.5.1, HK.3, BA.5.86, and JN.1 [13].

Furthermore, there has been little published evidence on long COVID persisting for more than two years [9], and this study helps to fill that gap. For children, lower rates of long COVID compared with adults have been documented in prior reviews [11,12], consistent with our findings. However, very few studies have followed children, including infants and young children, for more than two years [20,21], and this study contributes to the accumulation of evidence in this area.

With respect to symptom profiles, our findings were consistent with previous reports [7,8,22,23], showing that altered smell and altered taste were frequent during the Delta period, whereas cough became more common and smell/taste disturbances less frequent during the Omicron period. Importantly, this study further demonstrated that no distinctive changes in symptom patterns were observed in the most recent Omicron sublineages, which represents a novel contribution.

This study has several limitations. First, selection bias may have occurred, as participants with persistent symptoms may have been more likely to respond to the survey, potentially leading to an overestimation of overall prevalence. However, because data collection and analysis were conducted in a uniform manner across all groups, this bias is unlikely to have substantially affected comparisons between epidemic periods. Therefore, the absolute prevalence estimates reported in this study, including the 47% prevalence at six months during the Delta period, should be interpreted as potential upper limits rather than precise population-level estimates. Second, recall bias is possible, since the duration of symptoms was self-reported. To mitigate this, participants were asked to report duration in months, and interval-censored

survival analysis using the Turnbull method was applied to enhance reliability. In addition, interference with daily life was also self-reported, and its interpretation may have differed across age groups, particularly for pediatric participants assessed via proxy responses. Proxy-reported outcomes may underestimate the true prevalence of subjective symptoms—such as fatigue, difficulty concentrating, and sleep disturbances—that are not readily observable by caregivers. This limitation should be considered when interpreting the lower symptom prevalence and impact observed in children compared with adults. Third, genomic sequencing was not performed on individual patient samples to identify the infecting variants. To address this limitation, epidemic periods were classified based on publicly available variant surveillance data from Hiroshima Prefecture, which tracked the predominant circulating variants at the population level—a largely accepted practice in epidemiological studies of long COVID. Nevertheless, a lack of sequencing information for the infecting viral strain remains a caveat when interpreting the results. Such misclassification would likely bias estimates toward the null; therefore, the significant differences observed suggest that overestimation is unlikely. Fourth, the number of children infected during the wild-type and Alpha periods was small, making it difficult to conduct separate analyses of symptom characteristics by epidemic period in adults and children. In the supplementary analysis stratified by age group irrespective of epidemic period, cough and difficulty concentrating were relatively more frequent among children; however, this may partly reflect the predominance of Omicron-period infections in pediatric cases. Therefore, future studies should examine these age-specific and period-specific differences in greater detail. Fifth, data on vaccination history were available in our dataset; however, the association between vaccination and post-COVID-19 symptoms was already examined in our previous analysis using the same cohort [12], in which vaccination history was not significantly associated with symptoms persisting beyond three months in multivariable models. This finding does not imply that vaccination has no effect on long COVID in general, but rather may reflect characteristics of our cohort, such as the relatively small number of severe acute COVID-19 cases. Given that this association had already been addressed in our prior work, a further detailed evaluation of vaccination effects stratified by epidemic period was considered outside the scope of the present study, which primarily aimed to compare symptom trajectories across epidemic periods and between adults and children. Sixth, although the analysis was based on infection episodes recorded at the participating hospital, information on prior infections treated at other medical institutions was not fully available, particularly during later epidemic periods. Therefore, some participants may have experienced reinfections before the index episode analyzed in this study. While the survey design aimed to minimize misclassification by focusing on symptoms attributable to the specified infection episode, residual uncertainty regarding reinfections cannot be completely excluded.

In addition, information on co-infections or other intercurrent illnesses during the survey period was not systematically collected. Although the questionnaire explicitly asked respondents to report symptoms perceived as being caused by COVID-19 and persisting thereafter, attribution of symptoms based on self-report may not be perfect, and some degree of misclassification related to non–COVID-19 conditions cannot be completely excluded.

Seventh, this study was conducted at a single institution in Japan. However, this institution has served as a core facility for COVID-19 care in the region since the beginning of the pandemic and has managed a wide spectrum of patients, from children to older adults, suggesting reasonable representativeness of the local population. Eighth, the number of severe and critical cases was limited (critical cases: n = 3), and thus disease courses specific to severe cases could not be fully evaluated. Examining the relationship between acute treatment strategies and long-term symptom alleviation in critically ill patients is an important research question that should be addressed in future studies incorporating data from intensive care settings. Finally, all participants were Japanese, and differences in health care systems and access across countries may limit the generalizability of the findings.

In conclusion, this study demonstrated that the prevalence and duration of post-COVID-19 symptoms varied by epidemic period and between adults and children, with the highest risk observed during the Delta period and lower prevalence during the Omicron periods. The risk was particularly low among children aged ≤12 years. Nevertheless, persistent symptoms beyond two years were observed in approximately 20% of adults infected during the pre-Omicron periods,

10% during the Omicron periods, and in 4.1% and 1.9% of children infected during the Delta and Omicron-2022 periods, respectively. Importantly, our findings indicate that the frequency and duration of post-COVID-19 symptoms associated with recent Omicron sublineages circulating in 2024, including JN.1, were not substantially different from those observed during the earlier Omicron period, underscoring the current clinical relevance of this study.

These findings may also be relevant to other high-income settings with broadly accessible healthcare systems, although differences in healthcare structures and cultural contexts should be considered when interpreting the results. Taken together, these findings underscore the importance of continued long-term monitoring and the development of clinical and public health strategies that address the distinct risks associated with different variants and age groups.

## Supporting information

**S1 Table. Distribution of intervals for symptom resolution among participants with post-COVID-19 symptoms, used in Turnbull interval-censored survival analysis.**
(XLSX)

**S2 Table. Estimated prevalence of post-COVID-19 symptoms with 95% confidence intervals at selected time points, by epidemic periods.**
(XLSX)

**S1 Fig. Estimated prevalence of 13 post-COVID-19 symptoms at 3 and 12 months after infection, by age group.**
(PDF)

## Acknowledgments

We express our deep gratitude to all those who participated in this survey. Additionally, we are thankful for the cooperation of the Hiroshima City Hospital Organization, Hiroshima City Government, and Hiroshima Prefectural Government in conducting this survey.

## Author contributions

**Conceptualization:** Aya Sugiyama, Toshiro Takafuta, Tomoki Sato, Masao Kuwabara, Junko Tanaka.

**Data curation:** Aya Sugiyama.

**Formal analysis:** Aya Sugiyama, Tomoyuki Akita.

**Funding acquisition:** Aya Sugiyama, Junko Tanaka.

**Investigation:** Aya Sugiyama, Kanon Abe, Yayoi Yoshinaga, Junko Tanaka.

**Methodology:** Aya Sugiyama, Kanon Abe, Ko Ko, Tomoki Sato, Tomoyuki Akita, Junko Tanaka.

**Project administration:** Aya Sugiyama, Toshiro Takafuta, Junko Tanaka.

**Resources:** Toshiro Takafuta.

**Software:** Aya Sugiyama.

**Supervision:** Toshiro Takafuta, Ko Ko, Tomoyuki Akita, Masao Kuwabara, Shingo Fukuma, Junko Tanaka.

**Validation:** Tomoki Sato.

**Visualization:** Aya Sugiyama.

**Writing – original draft:** Aya Sugiyama.

**Writing – review & editing:** Toshiro Takafuta, Kanon Abe, Yayoi Yoshinaga, Ko Ko, Tomoki Sato, Tomoyuki Akita, Masao Kuwabara, Shingo Fukuma, Junko Tanaka.

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
