## [Decision Letter · Decision Letter 0]

21 Jan 2026

PONE-D-25-58216Differences in the Long-term Course of Post-COVID-19 Symptoms in Adults and Children across Epidemic Periods: A Retrospective Cohort Study in Japan, 2020–2024PLOS One

Dear Dr. Sugiyama,

Thank you for submitting your manuscript to PLOS ONE. After careful consideration, we feel that it has merit but does not fully meet PLOS ONE’s publication criteria as it currently stands. Therefore, we invite you to submit a revised version of the manuscript that addresses the points raised during the review process.

We look forward to receiving your revised manuscript.

Kind regards,

Rishi Jaiswal, Ph.D.

Academic Editor

PLOS One

Journal Requirements:

Additional Editor Comments:

Dear Dr. Sugiyama,

Thank you for submitting your manuscript entitled “Differences in the Long-term Course of Post-COVID-19 Symptoms in Adults and Children across Epidemic Periods: A Retrospective Cohort Study in Japan, 2020–2024” (Manuscript ID: PONE-D-25-58216) to PLOS ONE.

Your manuscript has now been evaluated by expert reviewers. The reviewers find the topic timely and important and acknowledge the strengths of your large retrospective cohort, extended follow-up period, and the comparative analysis across epidemic periods and age groups. However, they have raised several substantive concerns regarding the study design, data analysis, interpretation of results, and clarity of presentation that must be addressed before the manuscript can be considered further for publication.

Therefore, we invite you to revise your manuscript substantially and resubmit it as a Major Revision.

Reviewers' comments:

Reviewer's Responses to Questions

**Comments to the Author**

1. Is the manuscript technically sound, and do the data support the conclusions?

Reviewer #1: Partly

Reviewer #2: Yes

Reviewer #3: Yes

Reviewer #4: Yes

2. Has the statistical analysis been performed appropriately and rigorously? 

Reviewer #1: Yes

Reviewer #2: Yes

Reviewer #3: Yes

Reviewer #4: Yes

3. Have the authors made all data underlying the findings in their manuscript fully available?

Reviewer #1: No

Reviewer #2: Yes

Reviewer #3: Yes

Reviewer #4: Yes

4. Is the manuscript presented in an intelligible fashion and written in standard English?

Reviewer #1: Yes

Reviewer #2: Yes

Reviewer #3: Yes

Reviewer #4: Yes

5. Review Comments to the Author

Reviewer #1: This study with the title "Differences in the Long-term Course of Post-COVID-19 Symptoms in Adults and Children across Epidemic Periods: A Retrospective Cohort Study in Japan, 2020–2024" gives an insight about the long-term effects of Covid 19 and its co-relation with patient age. But the study seems to have a few unanswered questions. The authors need to discuss some of these questions: -

1. The authors discussed the different strains of SARS Covid-19 and their long-term symptoms in different patient groups. Do these patients get more than one strain or are they affected by multiple waves of the infection at different time points?

2. The study is based on a survey that includes children who can have different experiences than the adult involved. How did the authors provide more information and support to the children for specific responses.

3. In figure 2, it is interesting to find old versions of viruses showing more severity in symptoms than the later mutants arise. Is it consistent with time and people in each survey.

4. I did not find any information about the co-infection or any other illness that happened to the responders during the survey duration.

Reviewer #2: This study analyzes post-COVID-19 symptoms (PCC) in Japan across variants from Wild-type to Omicron in 2024. Including both adults and children fills a research gap. Using interval-censored survival analysis (Turnbull method) enhances confidence in duration estimates.

The authors note that vaccination history was not significantly associated with symptoms lasting more than three months in their earlier cohort analysis. However, they acknowledge that external studies (e.g., among U.S. veterans) observed a notable risk reduction with vaccination. Additional discussion of why this cohort might differ, such as the timing of vaccination relative to infection periods, especially during high-risk periods like the Delta variant, would be valuable.

The manuscript notes that although some children (1.9% to 4.1% depending on the period) experienced symptoms beyond two years, these did not disrupt daily life. Clarifying which symptoms, such as cough or difficulty concentrating, were most persistent would help create a clearer clinical picture.

The authors correctly point out that participants with ongoing symptoms might be more likely to respond. While they argue this does not skew comparisons across periods, they should explicitly note that the overall prevalence such as 47% at six months for Delta might represent an upper limit due to this potential bias.

The results indicate that newer sublineages (JN.1, etc.) are not significantly different from early Omicron variants (BA.1/2/5) in symptom duration. This is a novel finding and could be more prominently highlighted in the Conclusion to underscore the current relevance of the study.

Reviewer #3: This manuscript represents a strong and policy-relevant contribution to the long-COVID literature, particularly regarding variant-specific and age-specific trajectories. The suggested revisions are clarificatory rather than fundamental and can be addressed without additional analyses.

1. Figure 1: Flow diagram of the study population. Among all individuals diagnosed with COVID-19 at the participating hospital between March 2020 and July 2022 (N = 6,551), adults (n = 3,748) and children (n = 2,830) ; Adults and children number together isn’t added up to 6551.

Also, the final analysis population comprised 2,689 participants, including 1,524 adults and 1,165 children. These totals represent the combined analytic cohort derived from respondents to the initial survey (November 2022–March 2023) and newly diagnosed cases enrolled in 2024, However, the adult and pediatric totals do not correspond to a simple arithmetic sum of previously reported and newly recruited cases.

2. Explicitly clarify why vaccination was not included? It might really have an effect.

3. Clarify how “did not interfere with daily activities” was operationalized in children (parent-reported? school attendance?).

4. The conclusion could include one sentence emphasizing relevance to other high-income settings with similar healthcare access, while acknowledging cultural/system differences.

Reviewer #4: This is a well-conducted retrospective cohort study addressing an important and timely question regarding the long-term course of post-COVID-19 symptoms across epidemic periods and age groups. The long follow-up (extending beyond two years), inclusion of both adults and children within the same framework, and use of interval-censored survival analysis are clear strengths.

Major Comments:

1. Selection Bias: The study population consists of survey respondents, with response rates around 35–50% across waves.

2. Persistence Beyond Two Years: The manuscript states that symptoms persisting beyond two years showed “little further resolution.”

3. Variant Attribution: Epidemic periods were defined using regional surveillance data rather than individual-level viral sequencing. While this approach is appropriate, some parts of the Discussion imply variant-specific biological effects. I suggest framing the findings more consistently as period-based associations and clearly presenting variant-related explanations as hypotheses rather than causal conclusions.

4. Vaccination as an Unmeasured Factor: Vaccination status is not included in the current analysis, despite major differences in vaccine coverage across epidemic periods and age groups. The absence of vaccination data remains an important limitation in interpreting differences between Delta and Omicron periods. This should be more explicitly acknowledged, particularly when discussing faster symptom resolution during Omicron waves.

Minor Comments:

1. A brief justification for defining the outcome as “any symptom,” despite wide variation in severity and clinical impact, would be helpful.

2. The self-reported nature of “interference with daily life” may be interpreted differently across age groups; a short acknowledgment of this limitation would strengthen the methods.

3. Terminology (post-COVID-19 symptoms, post-COVID-19 condition, long COVID) could be used more consistently throughout the manuscript.

6. PLOS authors have the option to publish the peer review history of their article (what does this mean?). If published, this will include your full peer review and any attached files.

Reviewer #1: **Yes:**Rohit Tyagi

Reviewer #2: **Yes:**Arian Afzalian

Reviewer #3: No

Reviewer #4: No

---

## [Author Response · Author response to Decision Letter 1]

2 Feb 2026

Response to Reviewers

We sincerely thank the Academic Editor and the reviewers for their thorough evaluation of our manuscript and for the detailed and constructive comments provided. We appreciate their careful assessment of our study and their valuable suggestions, which have helped us to improve the clarity, rigor, and interpretation of the manuscript.

We have carefully considered all comments and have revised the manuscript accordingly. Below, we provide a point-by-point response to each comment. All changes made to the manuscript are indicated in the revised version with tracked changes.

Reviewer #1

This study with the title "Differences in the Long-term Course of Post-COVID-19 Symptoms in Adults and Children across Epidemic Periods: A Retrospective Cohort Study in Japan, 2020–2024" gives an insight about the long-term effects of Covid 19 and its co-relation with patient age. But the study seems to have a few unanswered questions. The authors need to discuss some of these questions: -

1. The authors discussed the different strains of SARS Covid-19 and their long-term symptoms in different patient groups. Do these patients get more than one strain or are they affected by multiple waves of the infection at different time points?

Response:

We thank the reviewer for this important and thoughtful comment. In this study, the analytic unit was the infection episode recorded at the participating hospital, which was treated as the index infection for the purposes of our analyses.

During the early phase of the COVID-19 pandemic in Japan, medical care for confirmed cases, including both outpatient visits and hospitalizations, was administratively coordinated by local governments. In Hiroshima, patients were largely concentrated at the participating hospital during this period. Therefore, for infections occurring in the early epidemic phases, it is reasonably likely that the recorded cases represent first infections.

In contrast, from 2022 onward, particularly during the Omicron-dominant periods, the rapid expansion of infections resulted in patients seeking care at a variety of medical institutions. Accordingly, we acknowledge the possibility that some individuals may have experienced a prior infection treated at another facility and subsequently visited the participating hospital for a later infection episode.

For participants included in the follow-up survey of the original cohort, we explicitly instructed respondents to report the duration of symptoms persisting from the previously surveyed infection episode and not to include symptoms newly developed after subsequent infections.

For the additional cohort of individuals infected during the Omicron-dominant period in 2024, the survey questions were structured to ask specifically about the timing of infection and symptoms associated with that infection episode. Thus, even if an individual had experienced a previous infection, the reported symptoms are expected to primarily reflect those associated with the Omicron 2024 infection.

Based on the reviewer’s suggestion, we have revised the Methods and Discussion sections as described below.

(Method) Page7, Line113-115

Among participants in the previous survey (November 2022–March 2023), 466 individuals (403 adults and 63 children) who still reported post-COVID-19 symptoms were followed up to collect additional information. For participants in the follow-up survey of the original cohort, respondents were explicitly instructed to report only symptoms persisting from the previously surveyed infection episode and not to include symptoms newly developed after subsequent infections. The current survey was conducted between December 2024 and March 2025, and the data obtained were integrated into the database established in the previous study for analysis.

(Discussion, Limitations) Page20, Line351-356

Sixth, although the analysis was based on infection episodes recorded at the participating hospital, information on prior infections treated at other medical institutions was not fully available, particularly during later epidemic periods. Therefore, some participants may have experienced reinfections before the index episode analyzed in this study. While the survey design aimed to minimize misclassification by focusing on symptoms attributable to the specified infection episode, residual uncertainty regarding reinfections cannot be completely excluded.

2. The study is based on a survey that includes children who can have different experiences than the adult involved. How did the authors provide more information and support to the children for specific responses.

Response:

We thank the reviewer for this important comment. In this study, responses for pediatric participants were obtained primarily through proxy responses provided by parents or legal guardians, which is a standard approach in pediatric survey research.

Parents or guardians were asked to complete the questionnaire based on their observations of the child’s symptoms and daily functioning. When children were able to communicate their symptoms, parents or guardians were allowed to consult with them while completing the survey. This approach was adopted to ensure accurate reporting while taking into account the child’s age, developmental stage, and ability to respond independently.

We have clarified this point in the Methods section of the revised manuscript.

(Method) Page7, Line125-128

For pediatric participants, questionnaire responses were obtained primarily through proxy responses provided by parents or legal guardians. Parents or guardians completed the survey based on their observations, with input from the child when appropriate.

3. In figure 2, it is interesting to find old versions of viruses showing more severity in symptoms than the later mutants arise. Is it consistent with time and people in each survey.

Response:

We thank the reviewer for this insightful comment. Figure 2 presents interval-censored survival curves generated using a consistent analytic framework (Turnbull method) across all epidemic periods, stratified by age group (adults and children). These curves are descriptive in nature and therefore reflect the observed distributions of symptom resolution, incorporating differences in participant characteristics, disease severity, and sample size across epidemic periods.

To account for these differences, we conducted adjusted analyses using Cox proportional hazards models. Importantly, even after adjustment for relevant covariates, the association between epidemic period and symptom resolution remained statistically significant.

We have clarified the descriptive role of Figure 2 and the complementary role of the adjusted Cox models in the Discussion section of the revised manuscript to avoid overinterpretation of variant-specific effects.

(Discussion) Page17, Line289-295

Before interpreting these differences, it should be noted that the comparisons across epidemic periods in this study represent period-based associations rather than direct causal effects of specific viral variants. Although the interval-censored survival curves were generated using a uniform analytic framework across all periods, they reflect underlying differences in participant composition, clinical severity, and sample size across epidemic periods. Importantly, however, the association between epidemic period and symptom resolution remained statistically significant after adjustment for relevant covariates in the Cox proportional hazards models.

4. I did not find any information about the co-infection or any other illness that happened to the responders during the survey duration.

Response:

We thank the reviewer for raising this important point. Information on co-infections or other intercurrent illnesses during the survey period was not systematically collected in this study.

However, the questionnaire was designed to specifically ask respondents to report symptoms that they perceived as having been caused by their COVID-19 infection and that persisted thereafter. Participants were instructed to focus on symptoms attributable to COVID-19 rather than newly developed symptoms due to other illnesses.

We acknowledge that self-reported attribution of symptoms may not be perfect and that some degree of misclassification cannot be entirely excluded. This limitation has been clarified in the Discussion section of the revised manuscript.

(Discussion, Limitations) Page20, Line357-361

In addition, information on co-infections or other intercurrent illnesses during the survey period was not systematically collected. Although the questionnaire explicitly asked respondents to report symptoms perceived as being caused by COVID-19 and persisting thereafter, attribution of symptoms based on self-report may not be perfect, and some degree of misclassification related to non–COVID-19 conditions cannot be completely excluded.

Reviewer #2

This study analyzes post-COVID-19 symptoms (PCC) in Japan across variants from Wild-type to Omicron in 2024. Including both adults and children fills a research gap. Using interval-censored survival analysis (Turnbull method) enhances confidence in duration estimates.

The authors note that vaccination history was not significantly associated with symptoms lasting more than three months in their earlier cohort analysis. However, they acknowledge that external studies (e.g., among U.S. veterans) observed a notable risk reduction with vaccination. Additional discussion of why this cohort might differ, such as the timing of vaccination relative to infection periods, especially during high-risk periods like the Delta variant, would be valuable.

Response:

We thank the reviewer for this important and constructive suggestion. In our earlier cohort analysis, vaccination history was not significantly associated with symptoms persisting beyond three months. One possible explanation, as discussed in that prior study, is that our study population included relatively few severe acute COVID-19 cases, which may have attenuated the observable protective effect of vaccination on long-term symptoms.

We have added a brief discussion of these points to the Discussion section of the revised manuscript.

(Discussion, Limitations) Page19-20, Line344-351

Fifth, the effects of vaccination were not directly evaluated in the present study. In our previous analysis using the same cohort(12), vaccination history was not significantly associated with post-COVID-19 symptoms persisting for more than three months after infection in multivariable models. One possible explanation is that the study population included relatively few severe acute COVID-19 cases, which may have limited the ability to detect a protective effect of vaccination on long-term outcomes. Although other studies have reported an association between vaccination and a reduced risk of long COVID in different settings (5), such an association was not observed in our cohort.

The manuscript notes that although some children (1.9% to 4.1% depending on the period) experienced symptoms beyond two years, these did not disrupt daily life. Clarifying which symptoms, such as cough or difficulty concentrating, were most persistent would help create a clearer clinical picture.

Response:

We thank the reviewer for this helpful and insightful comment. As suggested, we have clarified the clinical profile of symptoms persisting beyond two years among children. Specifically, we have added a brief description to the Results section indicating which symptoms remained in these cases, noting that they included fatigue, headache, dizziness, cough, difficulty concentrating, and sleep disorders, and that these symptoms did not interfere with daily activities.

(Results) Page13, Line232-233

Symptoms that persisted for more than 2 years showed little further resolution thereafter in both adults and children; however, in children, these symptoms did not interfere with daily activities and included fatigue, headache, dizziness, cough, difficulty concentrating, and sleep disorders.

The authors correctly point out that participants with ongoing symptoms might be more likely to respond. While they argue this does not skew comparisons across periods, they should explicitly note that the overall prevalence such as 47% at six months for Delta might represent an upper limit due to this potential bias.

Response:

We thank the reviewer for this important and helpful comment. As noted, individuals with persistent symptoms may have been more likely to respond to the survey, which could lead to an overestimation of the absolute prevalence of post-COVID-19 symptoms. While we believe that this response bias is unlikely to substantially affect comparisons across epidemic periods, we agree that the overall prevalence estimates, such as the 47% prevalence at six months during the Delta period, should be interpreted as potential upper limits rather than precise population-level estimates.

(Discussion, Limitations) Page19, Line326-328

First, selection bias may have occurred, as participants with persistent symptoms may have been more likely to respond to the survey, potentially leading to an overestimation of overall prevalence. However, because data collection and analysis were conducted in a uniform manner across all groups, this bias is unlikely to have substantially affected comparisons between epidemic periods. Therefore, the absolute prevalence estimates reported in this study, including the 47% prevalence at six months during the Delta period, should be interpreted as potential upper limits rather than precise population-level estimates.

We have clarified this point explicitly in the Discussion section of the revised manuscript.

The results indicate that newer sublineages (JN.1, etc.) are not significantly different from early Omicron variants (BA.1/2/5) in symptom duration. This is a novel finding and could be more prominently highlighted in the Conclusion to underscore the current relevance of the study.

Response:

We thank the reviewer for this insightful and encouraging comment. We agree that the finding that symptom duration associated with newer Omicron sublineages, including JN.1, did not substantially differ from that observed during the early Omicron period represents an important and timely contribution.

In response to this suggestion, we have revised the Conclusion section to more prominently highlight this finding and to emphasize the current relevance of our study.

(Discussion, Conclusion) Page21, Line375-378

Importantly, our findings indicate that the frequency and duration of post-COVID-19 symptoms associated with recent Omicron sublineages circulating in 2024, including JN.1, were not substantially different from those observed during the earlier Omicron period, underscoring the current clinical relevance of this study.

Reviewer #3

This manuscript represents a strong and policy-relevant contribution to the long-COVID literature, particularly regarding variant-specific and age-specific trajectories. The suggested revisions are clarificatory rather than fundamental and can be addressed without additional analyses.

1. Figure 1: Flow diagram of the study population. Among all individuals diagnosed with COVID-19 at the participating hospital between March 2020 and July 2022 (N = 6,551), adults (n = 3,748) and children (n = 2,830) ; Adults and children number together isn’t added up to 6551.

Also, the final analysis population comprised 2,689 participants, including 1,524 adults and 1,165 children. These totals represent the combined analytic cohort derived from respondents to the initial survey (November 2022–March 2023) and newly diagnosed cases enrolled in 2024, However, the adult and pediatric totals do not correspond to a simple arithmetic sum of previously reported and newly recruited cases.

Response:

We thank the reviewer for carefully identifying these inconsistencies in Figure 1.

First, we acknowledge that the numbers in the flow diagram were incorrectly reported in the original version. Specifically, among all individuals diagnosed with COVID-19 at the participating hospital between March 2020 and July 2022 (N = 6,551), the correct breakdown is adults (n = 3,748) and children (n

---

## [Decision Letter · Decision Letter 1]

3 Apr 2026

PONE-D-25-58216R1Differences in the Long-term Course of Post-COVID-19 Symptoms in Adults and Children across Epidemic Periods: A Retrospective Cohort Study in Japan, 2020–2024PLOS One

Dear Dr. Sugiyama,

Thank you for submitting your manuscript to PLOS ONE. After careful consideration, we feel that it has merit but does not fully meet PLOS ONE’s publication criteria as it currently stands. Therefore, we invite you to submit a revised version of the manuscript that addresses the points raised during the review process.

We look forward to receiving your revised manuscript.

Kind regards,

Rishi Kumar Jaiswal, Ph.D.

Academic Editor

PLOS ONE

Journal Requirements:

Additional Editor Comments :

Dear Dr. Sugiyama,

Thank you for submitting the revised version of your manuscript entitled “Differences in the Long-term Course of Post-COVID-19 Symptoms in Adults and Children across Epidemic Periods: A Retrospective Cohort Study in Japan, 2020–2024” to PLOS ONE.

The manuscript has now been evaluated based on the reviewers’ comments and your responses. I appreciate the efforts you have made to address the concerns raised during the previous round of review. The study is well-conducted and addresses an important topic.

However, a few minor issues remain that should be addressed before the manuscript can be considered for acceptance. I therefore invite you to submit a minor revision of your manuscript.

Please carefully address the reviewers’ remaining comments and provide a clear, point-by-point response outlining the changes made in the revised manuscript. Ensure that all revisions are clearly indicated.

Once these minor concerns are addressed, the manuscript is likely to be suitable for publication.

Thank you again for your submission to PLOS ONE. I look forward to receiving your revised manuscript.

Kind regards,

Rishi Kumar Jaiswal, Ph.D.

Academic Editor

PLOS ONE

Reviewers' comments:

Reviewer's Responses to Questions

**Comments to the Author**

1. If the authors have adequately addressed your comments raised in a previous round of review and you feel that this manuscript is now acceptable for publication, you may indicate that here to bypass the “Comments to the Author” section, enter your conflict of interest statement in the “Confidential to Editor” section, and submit your "Accept" recommendation.

Reviewer #1: All comments have been addressed

Reviewer #5: (No Response)

2. Is the manuscript technically sound, and do the data support the conclusions?

Reviewer #1: Yes

Reviewer #5: Yes

3. Has the statistical analysis been performed appropriately and rigorously? 

Reviewer #1: Yes

Reviewer #5: Yes

4. Have the authors made all data underlying the findings in their manuscript fully available?

Reviewer #1: Yes

Reviewer #5: Yes

5. Is the manuscript presented in an intelligible fashion and written in standard English?

Reviewer #1: Yes

Reviewer #5: Yes

6. Review Comments to the Author

Reviewer #1: "I would like to thank the authors for their detailed responses and the effort put into the revision. They have addressed all of my previous concerns satisfactorily. The manuscript is now much stronger and provides a valuable contribution to the field."

Reviewer #5: In the current work, the authors present a retrospective analysis of the persistence of COVID-19 associated symptoms (long COVID) in patients diagnosed with COVID-19 during different time periods which were dominated by different SARS-CoV-2 variants. The study provides a long term evaluation of the COVID-19 associated symptoms with comparison across the old and newer variants and also adult and children study groups and thus provides an important dataset from a healthcare and immunological standpoint. A few suggestions/comments regarding the study are as follows:

1. The study assigns different strains of SARS-CoV-2 to different years in which the patients were infected. While this is a largely accepted practice, a lack of sequencing information for the infecting viral strain still is a caveat while interpreting the study results.

2. In each of the category of patients, is there any information on the presence of vaccination and recovery from long COVID symptoms.

3. In the patients which require critical care, is there any co-relation between treatment strategies used and the better outcome in terms of long-term symptom alleviation.

4. In case of children as the reporting is done by the parents/supervising adults, the presence of symptoms which interfere with daily life might not be very accurate.

7. PLOS authors have the option to publish the peer review history of their article (what does this mean?). If published, this will include your full peer review and any attached files.

Reviewer #1: **Yes:**Rohit Tyagi

Reviewer #5: No

---

## [Author Response · Author response to Decision Letter 2]

7 Apr 2026

Response to Reviewers

We sincerely thank the Academic Editor and the reviewers for their careful evaluation of the revised manuscript and for the constructive comments provided in this second round of review. We appreciate their continued engagement with our work, which has helped us to further strengthen the manuscript.

We have carefully considered all remaining comments and have revised the manuscript accordingly. Below, we provide a point-by-point response to each comment. All changes made to the manuscript are indicated in the revised version with tracked changes.

Reviewer #1

I would like to thank the authors for their detailed responses and the effort put into the revision. They have addressed all of my previous concerns satisfactorily. The manuscript is now much stronger and provides a valuable contribution to the field.

Response:

We sincerely thank Reviewer #1 for the thorough and constructive review of our manuscript. We are grateful for the positive assessment and encouraging comments. The reviewer's thoughtful feedback in the previous round of review contributed greatly to strengthening the manuscript, and we deeply appreciate the time and effort devoted to this evaluation.

Reviewer #5

In the current work, the authors present a retrospective analysis of the persistence of COVID-19 associated symptoms (long COVID) in patients diagnosed with COVID-19 during different time periods which were dominated by different SARS-CoV-2 variants. The study provides a long term evaluation of the COVID-19 associated symptoms with comparison across the old and newer variants and also adult and children study groups and thus provides an important dataset from a healthcare and immunological standpoint. A few suggestions/comments regarding the study are as follows:

1. The study assigns different strains of SARS-CoV-2 to different years in which the patients were infected. While this is a largely accepted practice, a lack of sequencing information for the infecting viral strain still is a caveat while interpreting the study results.

Response:

We thank the reviewer for this comment. We agree that a lack of sequencing information for the infecting viral strain is a caveat when interpreting the results, as the reviewer noted. To address this limitation, epidemic periods were classified based on publicly available variant surveillance data from Hiroshima Prefecture, which tracked the predominant circulating variants at the population level—a largely accepted practice in epidemiological studies of long COVID. In response to the reviewer's comment, we have revised the Limitations section to explicitly acknowledge this caveat, while also noting that any resulting misclassification would likely bias estimates toward the null, making overestimation of variant-specific differences unlikely.

Original text

(Discussion, Limitations) Page 19, Line 332–334:

Third, genomic sequencing was not performed to identify the infecting variants, raising the possibility of some misclassification. Such misclassification would likely bias the results toward the null; therefore, the significant differences observed suggest that overestimation is unlikely.

Revised text

(Discussion, Limitations) Page 19, Line 335-342:

Third, genomic sequencing was not performed on individual patient samples to identify the infecting variants. To address this limitation, epidemic periods were classified based on publicly available variant surveillance data from Hiroshima Prefecture, which tracked the predominant circulating variants at the population level—a largely accepted practice in epidemiological studies of long COVID. Nevertheless, a lack of sequencing information for the infecting viral strain remains a caveat when interpreting the results. Such misclassification would likely bias estimates toward the null; therefore, the significant differences observed suggest that overestimation is unlikely.

2. In each of the category of patients, is there any information on the presence of vaccination and recovery from long COVID symptoms.

Response:

We thank the reviewer for this comment. Regarding vaccination history, data on vaccination status are available in our dataset. However, the association between vaccination history and post-COVID-19 symptoms was already examined and reported in our previous publication using the same cohort (reference 12). In that analysis, vaccination history was not significantly associated with post-COVID-19 symptoms persisting beyond three months in multivariable models. We wish to emphasize that this finding does not imply that vaccination has no effect on long COVID in general, but rather may reflect characteristics of our cohort, such as the relatively small number of severe acute COVID-19 cases. Given that the association between vaccination and long-term symptom outcomes had already been addressed in our prior work, and that the primary objective of the present study was to compare symptom trajectories across epidemic periods and between adults and children, a further detailed evaluation of vaccination effects was considered outside the scope of this study. We believe the Limitations section (Limitation 5) adequately addresses this point, but in response to the reviewer's comment, we have revised this section to more explicitly convey this rationale.

Original text

(Discussion, Limitations) Page 19-20, Line 340–347:

Fifth, the effects of vaccination were not directly evaluated in the present study. In our previous analysis using the same cohort (12), vaccination history was not significantly associated with post-COVID-19 symptoms persisting for more than three months after infection in multivariable models. One possible explanation is that the study population included relatively few severe acute COVID-19 cases, which may have limited the ability to detect a protective effect of vaccination on long-term outcomes. Although other studies have reported an association between vaccination and a reduced risk of long COVID in different settings (5), such an association was not observed in our cohort.

Revised text

(Discussion, Limitations) Page 20, Line 348-357:

Fifth, data on vaccination history were available in our dataset; however, the association between vaccination and post-COVID-19 symptoms was already examined in our previous analysis using the same cohort (12), in which vaccination history was not significantly associated with symptoms persisting beyond three months in multivariable models. This finding does not imply that vaccination has no effect on long COVID in general, but rather may reflect characteristics of our cohort, such as the relatively small number of severe acute COVID-19 cases. Given that this association had already been addressed in our prior work, a further detailed evaluation of vaccination effects stratified by epidemic period was considered outside the scope of the present study, which primarily aimed to compare symptom trajectories across epidemic periods and between adults and children.

3. In the patients which require critical care, is there any co-relation between treatment strategies used and the better outcome in terms of long-term symptom alleviation.

Response:

We thank the reviewer for raising this clinically important question. In the present study, the number of severe and critical cases was extremely limited (critical cases: n=3), precluding any meaningful analysis of the relationship between treatment strategies and long-term outcomes in this subgroup. In response to the reviewer's comment, we have expanded the relevant Limitations section to explicitly acknowledge this issue and to identify it as an important direction for future research.

Original text

(Discussion, Limitations) Page 20, Line 362–364:

Eighth, the number of severe cases was limited, and thus disease courses specific to severe cases could not be fully evaluated. This issue should be addressed in other cohorts.

Revised text

(Discussion, Limitations) Page 21, Line 372-376:

Eighth, the number of severe and critical cases was limited (critical cases: n=3), and thus disease courses specific to severe cases could not be fully evaluated. Examining the relationship between acute treatment strategies and long-term symptom alleviation in critically ill patients is an important research question that should be addressed in future studies incorporating data from intensive care settings.

4. In case of children as the reporting is done by the parents/supervising adults, the presence of symptoms which interfere with daily life might not be very accurate.

Response:

We thank the reviewer for this important methodological point. We fully agree that proxy-reported outcomes in children may not accurately capture the presence or severity of symptoms that interfere with daily life, particularly for subjective symptoms—such as fatigue, difficulty concentrating, and sleep disturbances—that are not readily observable by caregivers. In response to the reviewer's comment, we have expanded the relevant Limitations section to more explicitly discuss the direction and potential impact of this bias.

Original text

(Discussion, Limitations) Page 19, Line 330–331:

In addition, interference with daily life was also self-reported, and its interpretation may have differed across age groups, particularly for pediatric participants assessed via proxy responses.

Revised text

(Discussion, Limitations) Page 19, Line 332-335:

In addition, interference with daily life was also self-reported, and its interpretation may have differed across age groups, particularly for pediatric participants assessed via proxy responses. Proxy-reported outcomes may underestimate the true prevalence of subjective symptoms—such as fatigue, difficulty concentrating, and sleep disturbances—that are not readily observable by caregivers. This limitation should be considered when interpreting the lower symptom prevalence and impact observed in children compared with adults.

---

## [Editor Report · Decision Letter 2]

24 Apr 2026

Dear Dr. Sugiyama,

I hope you are doing well.

I am pleased to inform you that your manuscript entitled *“Differences in the Long-term Course of Post-COVID-19 Symptoms in Adults and Children across Epidemic Periods: A Retrospective Cohort Study in Japan, 2020–2024”* (Manuscript Number: PONE-D-25-58216R2) has been **accepted for publication** in PLOS ONE.

The reviewers and editorial team appreciate the thorough revisions you have made, which have significantly strengthened the manuscript. Your study provides valuable insights into the long-term trajectory of post-COVID-19 symptoms across different epidemic periods and age groups, and will be of considerable interest to the scientific and clinical community.

The manuscript will now proceed to the production stage. You will be contacted by the journal’s production team regarding the next steps, including proof review.

Thank you for submitting your work to PLOS ONE. We look forward to your future contributions.

With best regards,

Dr. Rishi Kumar Jaiswal

Academic Editor

PLOS ONE

---

## [Editor Report · Acceptance letter]

PONE-D-25-58216R2

PLOS One

Dear Dr. Sugiyama,

I'm pleased to inform you that your manuscript has been deemed suitable for publication in PLOS One. Congratulations! Your manuscript is now being handed over to our production team.

Kind regards,

on behalf of

Dr. Rishi Jaiswal

Academic Editor

PLOS One